# Exploring the Antimicrobial, Antioxidant, and Antiviral Potential of Eco-Friendly Synthesized Silver Nanoparticles Using Leaf Aqueous Extract of *Portulaca oleracea* L.

**DOI:** 10.3390/ph17030317

**Published:** 2024-02-28

**Authors:** Mohammed Ali Abdel-Rahman, Khalid S. Alshallash, Ahmed M. Eid, Saad El-Din Hassan, Mutaz Salih, Mohammed F. Hamza, Amr Fouda

**Affiliations:** 1Botany and Microbiology Department, Faculty of Science, Al-Azhar University, Nasr City, Cairo 11884, Egypt; mohamedali@azhar.edu.eg (M.A.A.-R.); aeidmicrobiology@azhar.edu.eg (A.M.E.); saad_hassan@azhar.edu.eg (S.E.-D.H.); 2Department of Biology, College of Science, Imam Mohammad Ibn Saud Islamic University (IMSIU), Riyadh 11623, Saudi Arabia; 3Department of Chemistry, College of Science, Imam Mohammad Ibn Saud Islamic University (IMSIU), Riyadh 11623, Saudi Arabia; meamin@imamu.edu.sa; 4School of Nuclear Science and Technology, University of South China, Hengyang 421001, China; 5Nuclear Materials Authority, P.O. Box 530, El-Maadi, Cairo 11728, Egypt

**Keywords:** green synthesis, *Portulaca oleracea*, silver nanoparticles, biomedical applications, antimicrobial, antiviral activity

## Abstract

Herein, the prospective applications of green fabricated silver nanoparticles (Ag-NPs) within the biomedical field were investigated. The leaf aqueous extract of *Portulaca oleracea* L., a safe, cheap, and green method, was used to fabricate Ag-NPs. The maximum plasmon resonance of synthesized NPs has appeared at 420 nm. The various biomolecules present in the plant extract to assemble spherical Ag-NPs with sizes of 5–40 nm were analyzed using Fourier transform infrared and transmission electron microscopy. The Ag was the major content of the formed Ag-NPs with an atomic percent of 54.95% and weight percent of 65.86%, as indicated by EDX. The crystallographic structure of synthesized NPs was confirmed by the diffraction of the X-ray. The dynamic light scattering exhibits the homogeneity and mono-dispersity nature with a polydispersity index of 0.37 in the colloidal fluid and a zeta potential value of –36 mV. The synthesized Ag-NPs exhibited promising antimicrobial efficacy toward various prokaryotic and eukaryotic pathogenic microorganisms with low MIC values of 12.5 µg mL^−1^ and 6.25 µg mL^−1^, respectively. Additionally, the *P. oleracea*-formed Ag-NPs showed optimistic antioxidant activity assessed by DPPH and H_2_O_2_ assay methods with the highest scavenging percentages of 88.5 ± 2.3% and 76.5 ± 1.7%, respectively, at a concentration of 200 µg mL^−1^. Finally, the biosynthesized Ag-NPs showed high antiviral properties toward the hepatitis A virus and Cox-B4 with inhibition percentages of 79.16 ± 0.5% and 73.59 ± 0.8%, respectively. Overall, additional research is essential to explore the Ag-NP-based aqueous extract of *P. oleracea* for human health. In the current investigation the use of synthesized Ag-NPs as antimicrobial, antioxidant, and antiviral agents to protect against pathogenic microbes, degenerative diseases caused by various oxidative stresses, and deadly viruses is recommended.

## 1. Introduction

Nanobiotechnology has appeared as an interesting study area for different researchers due to its wide applications in varied sectors such as agriculture, wastewater treatment, catalysis, cloth manufacture, optical and electrical sensors, cosmetics, drug delivery, and medications [1,2]. Nanoparticles (NPs) have unique features such as small sizes (1 nm to 100 nm), thermal conductivity, shapes, plasmonic and magnetic characteristics, and a high surface area to volume [3,4] as compared to counter-bulk materials. Generally, these new active substances are produced by varied methods including chemical, physical, and biological ones. Unfortunately, those synthesized using chemical and physical methods are non-ecofriendly, have a low yield, require harsh environmental conditions, produce toxic by-products, [5,6]. Due to all these drawbacks, the use of NPs formed using chemical and physical methods is not preferred to be applied in medical sectors. Thus, there is an immediate need for methods that are non-toxic, cheap, and environmentally benign for synthesizing NPs. The green synthesis or biosynthesis methods use of natural products from plants or microorganisms, such as yeast, fungi, actinomycetes, bacteria, and algae, to reduce the metal precursor to form nanoscale substances [7,8]. Therefore, environmental friendliness, reduced environmental effect, and the possibility of creating NPs with improved biocompatibility are the reasons for the growing interest in green synthesis processes [9,10].

In this regard, the huge amount of metabolites secreted by the plant are used as reducing agents to form a wide range of NPs such as Au (gold), Ag (silver), Se (selenium), palladium (Pd), ZnO (zinc oxide), CuO (copper oxide) Fe (iron oxide), etc. Plant metabolites, such as phenols, proteins, carbohydrates, alkaloids, flavonoids, terpenoids, and polysaccharides, have dual roles as reducing metal ions to assemble NPs and coating the final nanostructure substances to increase their stability [11]. *Portulaca oleracea*, also called Purslane or Regla in Egypt, belongs to the portulacaceae family. Several reasons distinguished *P. oleracea* from other plant extracts, which led to its selection for the form of silver nanoparticles (Ag-NPs). Among these reasons, *P. oleracea* is well-known for its abundant phytochemical makeup, which includes polysaccharides, omega-3, flavonoids, vitamins (A, B, and C), alkaloids, minerals (Mg, Ca, K, and Fe), and terpenoids [12,13]. These compounds are thought to have powerful reducing and capping capabilities that are ideal for synthesizing nanoparticles. Moreover, *P. oleracea* is extensively spread and readily available. In addition, this plant is characterized by its medicinal importance, suggesting that synthesized Ag-NPs using their aqueous extract will have a wide range of biomedical applications. Due to these reasons, the aqueous extract of *P. oleracea* leaves is used to assemble varied NPs with specific sizes and shapes. For instance, the *P. oleracea*-aqueous extract was used to produce CuO-NPs which revealed promising activities involving antimicrobial, wastewater treatment, and the removal of heavy metals [14]. Additionally, *P. oleracea* was used to synthesize Ag-NPs and ZnO-NPs with anticancer efficacy [15]. Although there have been several investigations to produce Ag-NPs using different plants, the use of *P. oleracea* leaf extract for this objective has received very little attention to fill the gap between the medicinal characteristic of this plant and their efficacy to green synthesis of Ag-NPs.

The Ag-NPs have attracted the researcher’s attention due to significant input in various medical, scientific, and industrial fields because of their unique characteristics. Ag-NPs are recognized as non-toxic to humans and extremely toxic to microbes such as bacteria and fungi [16]. In addition to antimicrobial activity, Ag-NPs have a wide range of applications such as wound healing, catalysis, optical properties, coating and conductive ink, drug delivery, cancer therapy, water and air purification, food packaging, diabetic treatment, cloth industry, cosmetics, and photothermal therapy [16,17]. Various published investigations have recorded the efficacy of plant extract to assembly the Ag-NPs, for instance, *Phaseolus vulgaris* [18], *Eugenia roxburghii* [7], *Calophyllum tomentosum* [17], *Malephora lutea* [19], *Crossopteryx febrifuga* [20], *Erythrina suberosa* [21], *Atrocarpus altilis* [22], *Prosopis farcta* [11], and *Psidium guajava* [23]. The synthesized Ag-NPs using plant extracts showed higher activities than those formed using other biological entities due to the varied capping agents that coated NPs and originated from plant extract [24]. In addition to the advantages of green synthesized methods compared to conventional approaches, our study fills a gap in the literature through the development and utilization of *P. oleracea*-synthesized Ag-NPs in various applications, including antimicrobial, antioxidant, and antiviral activities, which have significant implications for healthcare and environmental protection.

The purpose of this study is to contribute to the expanding body of knowledge on green nanotechnology and its participation in the healthcare and environmental sustainability. Following this, the Ag-NPs were synthesized by water extract of the *P. oleracea* and characterized using several analytical tools, including transmission electron microscopy, energy-dispersive ray, X-ray pattern, zeta potential, and UV-vis spectroscopy. The activity of Ag-NPs as an antimicrobial agent toward prokaryotic and eukaryotic pathogenic microorganisms, antioxidant compounds (assessed by DPPH and H_2_O_2_ assay method), and their effectiveness against HAV and Cox-B4 viruses were investigated.

## 2. Results and Discussion

### 2.1. Portulaca oleracea-Mediated Biosynthesis of Ag-NPs

Recently, the plant has attracted a lot of attention in the nanotechnology field due to the unusual qualities that it possesses, particularly in the biosynthesis of Ag-NPs [25]. This green synthesis method involves the utilization of plant extracts to convert silver ions into nanostructures that are both stable and biocompatible. Herein, the *P. oleracea* water extract was used in the environmentally friendly manufacture of NPs, and offers several advantages over traditional chemical approaches. One of the most important advantages is that the method is low impact on the environment. In contrast to chemical processes, which frequently involve the use of dangerous chemicals and result in the production of deleterious by-products [14]. Phenolic compounds in addition to flavonoids, alkaloids, and terpenoids are only some of the bioactive chemicals that are abundant in *P. oleracea* [26]. These bioactive compounds facilitate the reduction of Ag ions into Ag-NPs. In addition, the phytochemicals perform the function of capping agents, which prevents the NPs from aggregating and ensures NP stability. Ag-NPs with improved biocompatibility are produced as a result of this green synthesis method, which is not only beneficial to the atmosphere but also environmentally benign. In a similar study, various NPs were fabricated through the reducing action of active metabolites that exist in the water extract of the plant leaves [27,28].

### 2.2. Characterization

#### 2.2.1. Optical Properties Using UV-Vis Spectroscopy

The change from colorless to yellowish-brown visually confirms the successful formation of Ag-NPs using the *P. oleracea* water extract. The shifting of the color can be attributed to the excitation of surface plasmon vibrations in the Ag-NPs, which is responsible for this phenomenon. Surface plasmon resonance (SPR) is a phenomenon that is defined by the collective oscillation of free electrons on the surface of a metal when it is exposed to light [29]. In order to verify the synthesis of NPs and determine their optical characteristics, particularly the maximum SPR, UV-vis spectroscopy is a technique that is commonly utilized. In this investigation, the UV-vis spectrum of *P. oleracea*-mediated Ag-NPs exhibited a prominent absorption peak at 420 nm (Figure 1A). This wavelength corresponds to the maximum absorption of light by the Ag-NPs during the SPR. Similarly, the Ag-NPs produced by *P. oleracea* exhibited their highest specific photo-response (SPR) at a 423 nm wavelength [28]. Dong and coauthors reported that the greenly produced Ag-NPs have a localized SPR in the 400 to 460 nm range [30].

The particular wavelength is related to the electron oscillation frequency on the NPs’ surface, which is affected by their size, shape, and environmental factors [31]. A stable colloidal solution and nanoparticles with consistent sizes and shapes are indicated by the sharp and clearly defined peak. According to Taha et al., spherical Ag-NPs often exhibit a maximum SPR peak in the 410–420 nm range [32].

#### 2.2.2. Fourier Transform Infrared (FT-IR)

Fourier transform infrared spectroscopy (FT-IR) is a common method for identifying the functional groups in plant aqueous extracts and how they contribute to NP production. In addition, FT-IR was able to identify biosynthesized NPs with new functional groups or those with shifted presence groups. The *P. oleracea* water extract contains six peaks at 3420, 2080, 1643, 1430, 1290, 720, and 520 cm^−1^ (Figure 1B). These peaks were shifted or formed new peaks after the fabrication of Ag-NPs. The peak at 3420 cm^−1^ was signified by the O-H vibration stretching or N-H groups. These groups are associated with the different compounds such as sugars, phenolic compounds, or organic molecules present in plant aqueous extract [33]. The intensity of this peak decreased as well as their position was shifted at 3427 cm^−1^ after the Ag-NPs form. Moreover, the peak at 2080 cm^−1^ refers to the stretching C=C due to the existence of alkynes or other compounds containing triple pond in plant aqueous extracts [34]. The peak localized at 1643 cm^−1^ is related to the C=O (carbonyl group) stretching of amides or ketones or refers to the N-H bending of secondary amines [35]. After the production of NPs, this peak was moved to a frequency of 1624 cm^−1^. The peak at a frequency of 1430 cm^−1^ is linked to the bending C-H vibration of alkanes or methyl groups. These groups indicate the presence of organic compounds containing CH_3_ such as certain types of carbohydrates or lipids [36]. The peak at 1290 cm^−1^ is related to the stretching of C-O that is associated with carbohydrates and polysaccharides [37]. The peaks at 720 and 520 cm^−1^ are associated with the out-of-plane bending of the aromatic ring in plant aqueous extract [38]. Because of the manufacturing of Ag-NPs, these peaks moved to the 825 and 777 cm^−1^ frequency range. The presence of new peaks due to Ag-NPs formation was observed at the wavenumbers of 2924, 1383, and 1120 cm^−1^. The peaks at 2924 and 1383 cm^−1^ signify the stretching vibration of C-H in aliphatic compounds or organic molecules that act as a capping or stabilizing agent during Ag-NPs synthesis [39,40]. Finally, the peak at 1120 cm^−1^ is related to stretching C-O associated with molecules containing oxygen that have a role in the reduction and stabilization of as-formed Ag-NPs [41]. The FTIR spectra, in general, revealed functional groups that are indicative of the participation of plant-based chemicals in the green synthesis process. More specifically, these chemicals were associated with the reduction and stability of Ag-NPs.

#### 2.2.3. Morphological and Elemental Composition Investigation

The physicochemical characterization of Ag-NPs is an important tool for understanding and optimizing their biological properties. To reveal the size, shape, and elemental composition of Ag-NPs, transmission electron microscopy (TEM) and energy-dispersive X-ray spectroscopy (EDX) are crucial instruments. In the present investigation, it was found that the water extract of *P. oleracea* leaves possesses the capability to produce spherical Ag-NPs with diameters ranging from 5 to 40 nm, with an average particle size of 13.75 ± 8.7 nm (as depicted in Figure 2A,B). Furthermore, as shown in Figure 2A, the TEM investigation shows that the green-produced NPs are neatly distributed and free of aggregation. Similarly, using TEM, the shape and size of the Ag-NPs that were produced by aqueous extract of *Eugenia roxburghii* are spherical, with sizes ranging from 19 to 39 nm with an average of 24 nm [7]. Furthermore, the shape of synthesized Ag-NPs using cell-free supernatant of *Aspergillus flavus* was spherical with a size of 12.5 based on the TEM image [42].

The detection of sizes and shapes of Ag-NPs is an important analysis due to the association between these parameters and biological activity. The surface area and reactivity are both affected by size, while the exposed facets are determined by form. Both of these factors have a substantial impact on the interactions that occur with biological organisms [43]. This finding was consistent with the findings that were recorded by Kambale et al. [20]. Those authors were able to fabricate Ag-NPs using the aqueous extract for three plants, *Senna siamea*, *Crossopteryx febrifuga*, and *Brillantaisia patula*, with sizes of 115 nm, 47 nm, and 45 nm, respectively, based on TEM analysis. *Staphylococcus aureus*, *Pseudomonas aeruginosa*, and *Escherichia coli* are the three bacteria that the authors analyze in order to determine the antibacterial activity of three different Ag-NPs that were created. The data showed that the Ag-NPs formed using an aqueous extract of *B. patula* showed promising antibacterial activity, which was greater than those recorded by *C. febrifuga* and *S. siamea*. The authors suggested that this activity could be related to the smaller size of Ag-NPs formed by *B. patula* than others. In addition, this activity could be attributed to the capping agents which varied between plants. On the other hand, Acharya and co-authors successfully formed spherical and rod-shaped Ag-NPs and investigated their antibacterial activity against different Gram-positive (G+ve) strains and Gram-negative (G-ve) strains [44]. Interestingly, the activity of the spherical shape against pathogenic bacterial strains was greater than the activity recorded by the rod shape.

The EDX analysis was utilized to evaluate the composition of the Ag-NPs that were produced through plant-mediated biosynthesis. As shown, identical absorption peaks were observed at bending energies of 2.8, 3.0, and 3.2 KeV [20]. The EDX analysis reveals that the silver peaks were the most dominant in the sample with weight and atomic percentages of 65.86% and 54.95%, respectively (Figure 2C). The absence of an N peak indicates that the activity of plant aqueous extract reduces all metal precursors (AgNO_3_) and forms metallic Ag-NPs. One possible explanation for the appearance of other peaks, such as O, C, and Cl, is that they are caused by the scattering of capping agents that coat the produced Ag-NPs [45].

#### 2.2.4. X-ray Diffraction (XRD)

Analysis using XRD is an essential method for determining the characteristics of Ag-NPs and acquiring an understanding of the structural properties of these particles. At two theta values of 38.14°, 44.19°, 64.45°, and 77.37°, Figure 2D displays four diffraction peaks. These peaks correspond to the Bragg’s reflection of (111), (200), (220), and (311), respectively. In accordance with the standard of JCPDS-file No-04-0783, the XRD patterns that were obtained are consistent with the face-centered cubic (FCC) structure of metallic Ag-NPs [20]. These data confirmed the green synthesized Ag-NPs were crystalline in nature [46]. Consistent with the results of the EDX investigation, the presence of extra XRD peaks suggests the presence of impurities originating from the plant extract. The crystallite size of Ag-NPs was determined by employing the Debye–Scherrer equation, which was derived from the XRD examination. The analysis of the data revealed that the typical size of the crystallites was 38 nm. The collected data are consistent with those that determine the average crystallite size of Ag-NPs manufactured by the aqueous extract of *Eugenia roxburghii* to be 35 nm. This calculation is based on the Debye–Scherrer formula [7].

#### 2.2.5. Dynamic Light Scattering (DLS)

DLS analysis indicated that the biosynthesized Ag-NPs in the colloidal fluid had a diameter of 83.5 nm (Figure 3A). As shown, the obtained Ag-NP diameter by DLS was greater than the sizes obtained by TEM and XRD. Similarly, the particle sizes of formed Ag-NPs by the water extract of *Phaseolus vulgaris* were in the ranges of 12–15 nm as detected by TEM, whereas it increased to 188 nm after detection by DLS analysis [18]. These differences could be attributed to a wide range of factors. For instance, the calculation of particle size using DLS is affected by capping agents that originated from plant extract and coated the NPs. Additionally, the DLS measures the hydrodynamic residue of NPs in the hydrated state whereas the TEM measures the particle size as a solid state. Furthermore, the effect of the colloidal fluid’s non-uniform distribution of produced nanoparticles on DLS observations is another reason why TEM and DLS computations differ [47].

The mono-dispersity or polydispersity of Ag-NPs in the colloidal fluid was detected using DLS analysis by measuring the polydispersity index (PDI). Honary et al. reported that the particles were described as polydisperse and monodisperse when the PDI value was greater or less than 0.7, respectively [48]. In this investigation, the produced Ag-NPs have a PDI value of 0.37 which indicates the mono-dispersity nature. The obtained results were matched with those reported that the PDI value of Ag-NPs synthesized by *Eucalyptus camaldulensis* was 0.399 which confirmed the mono-dispersity nature [25]. The utilization of monodisperse silver nanoparticles demonstrated superior performance and novel applications in comparison to their polydisperse counterparts.

#### 2.2.6. Zeta Potential (ζ)

More investigation about the NPs’ stability and surface charge were assessed using ζ-potential analysis. Based on a ζ-potential study, the following information was given as the criteria for classifying the stability of nanoparticles (NPs): ±0–10 mV is a range that correlates to highly unstable particles, whereas ±10–20 mV is considered to be generally stable. On the other hand, ±20–30 mV is considered to be moderate, whereas a value greater than ± 30 mV is considered to be highly stable, respectively [49]. The ζ-value of the Ag-NPs derived from *P. oleracea* plants was −36 mV (Figure 3B), demonstrating that the synthesized NPs are highly stable. In a related work, the ζ-potential of Ag-NPs produced by a water leaves’ extract of *Eugenia roxburghii* was found to be −37.8 mV, indicating excellent stability [7]. Capping agents, such as polyphenolic and flavonoid compounds found in water-based extracts, may be responsible for the Ag-NPs surface’s negative charge [7]. Curiously, the absence of agglomeration or aggregation over time is prevented by the presence of a single charge, a negative charge, on the surface of the NPs.

### 2.3. Antimicrobial Activity

The agar well diffusion method was used to assess the effectiveness of manufactured Ag-NPs in inhibiting the development of several pathogenic microorganisms, such as *Staphylococcus aureus*, *Bacillus subtilis*, *Pseudomonas aeruginosa*, *Escherichia coli*, *Candida albicans*, and *Aspergillus brasiliensis*. Results showed that the action of Ag-NPs against several bacteria was concentration-dependent. Consistent with previously published research, Ag-NPs synthesized using aqueous *Psidium guajava* leaf extract had a concentration-dependent antimicrobial effect against a variety of bacteria, including *Bacillus megaterium*, *B. subtilis*, *Arthrobacter creatinolyticus*, *E. coli*, *Alcaligenes faecalis*, *Saccharomyces cerevisiae*, *Aspergillus niger*, *Acinetobacter baumannii*, and *Rhizopus oryzae* [23]. Additionally, the antimicrobial efficacy of produced Ag-NPs using fruit water extract of *Prosopis farcta* against *Streptococcus pneumonia*, *Staphylococcus aureus*, *Salmonella typhi*, and *E. coli* was dose-dependent [11].

In this investigation, the topmost activity was verified for a concentration of 100 µg mL^−1^ with clear zones of 20.7 ± 0.5, 18.7 ± 0.7, 15.3 ± 0.5, 17.4 ± 1.02, 16.3 ± 0.5, and 15.5 ± 0.6 mm for *P. aeruginosa*, *E. coli*, *B. subtilis*, *S. aureus*, *C. albicans*, and *A. brasiliensis*, respectively (*p* ≤ 0.001) (Figure 4). At low concentrations, the diameter of these transparent zones diminished. Clear zones measuring 9.3 ± 0.5, 12.5 ± 0.6, 13.7 ± 0.6, 11.7 ± 0.5, 11.0 ± 0.5, and 10.3 ± 0.6 mm were reduced for the same sequence of microorganisms when the concentration was 12.5 µg mL^−1^. It is worth noting that for G-ve and *S. aureus* (G+ve), the activity of the produced Ag-NPs remained at a low concentration (6.25 µg mL^−1^). Likewise, the Ag-NPs produced by the callus aqueous extract of *Solanum incanum* exhibited their maximum antimicrobial activity at a concentration of 200 µg mL^−1^, inhibiting the growth of *S. aureus*, *B. subtilis*, *Klebsiella pneumoniae*, *E. coli*, *P. aeruginosa*, and *C. albicans* with zone inhibition values ranging from 19.8 to 24.2 mm. The minimum activity was achieved at a concentration of 12.5 µg mL^−1^ [50].

The minimum inhibitory concentration (MIC) value is a critical point that should be detected before integrating the Ag-NPs into medication. Herein, synthesized Ag-NPs had a MIC value of 12.5 µg mL^−1^ against *B. subtilis* and fungi, with clear zones ranging from 9.3 to 11 mm, and 6.25 µg mL^−1^ against *S. aureus* and G-ve (*P. aeruginosa* and *E. coli*), with clear zones ranging from 9.7 to 11.3 mm (Table 1). The collected data were compared to the MIC of Ag-NPs produced by *Streptomyces antimycoticus*, which was 25 µg mL^−1^ for *B. subtilis* and *E. coli* and 12.5 µg mL^−1^ for *S. aureus* and *Salmonella typhi* [51].

It was found that the produced Ag-NPs had better activity against G-ve bacteria than G+ve. Bacterial cell walls differ in structure, which is associated with this action. G+ve bacteria are unable to absorb Ag-NPs because their cell walls are thick with peptidoglycan. In contrast, G-ve bacteria cell walls contain just thin coatings of peptidoglycan, which allows them to facilitate NP ingress and, ultimately, renders nearly all intracellular macromolecules inoperable [25]. This phenomenon is considered one of the antibacterial mechanisms of NPs. Furthermore, the superior antibacterial activity of Ag-NPs toward G-ve bacteria could be related to the presence of more lipopolysaccharides that increase the negatively charged on the surface of G-ve bacteria and hence enhance the electrostatic attraction with Ag-NPs that are positively charged [52]. Moreover, the efflux pump that is responsible for the removal of harmful compounds from the inside of the cells may differ in G-ve bacteria compared to G+ve. The green formed Ag-NPs can overcome the efflux pump mechanism in G-ve bacteria, leading to accumulation inside the cells and hence enhance their antibacterial activity [53].

Another antimicrobial mechanism is the release of Ag^+^ ions inside the cells upon Ag-NP entrance. Reactive oxygen species (ROS) are produced more readily as a result, which can harm the respiratory system of cells and cause cell death [30]. Furthermore, interacting with NPs damages the cytoplasmic membrane’s selective permeability function [1]. Ag-NPs can rupture the sterol that is present in the cell wall of *Candida* by blocking the ergosterol pathway synthesis leading death of the cells [54]. The activity of Ag-NPs against multicellular fungi could be attributed to various mechanisms such as the detachment of the toxic ions (Ag^+^) and their deleterious impact on macromolecules and cytoplasmic membrane function, inhibition of the germination of conidia due to the reaction of NPs with it, disrupt the chain of electron transport as a result of dysfunction of selective permeability function, and destroy the mitochondria due to high production of ROS (oxidative stress) [1].

### 2.4. Antioxidant Activity

Discovery novel substances exhibiting antioxidant action is of utmost significance in the health and medicine sectors. Cancer, heart disease, and neurological disorders are just a few of the many diseases that free radicals can cause [55]. Therefore, finding new antioxidant molecules is vital in the development of new treatments and new strategies to prevent these diseases. NPs are candidates as promising antioxidant compounds due to their uncommon physicochemical features, such as small sizes and the ratio between their surface and volume, which improve their activity and enhance their interaction with biological systems to scavenge free radicals [32,56]. In this investigation, two methods, namely DPPH and H_2_O_2_ assay methods, were used to evaluate the efficacy of green synthesized Ag-NPs in scavenging the free radicals (Figure 5), the antioxidant capacity of Ag-NPs was concentration-dependent, the activity was increased with high NP concentration and vice versa [17].

Using the DPPH assay method, the highest scavenging activity was recorded for 200 µg mL^−1^ of Ag-NPs with percentages of 88.5 ± 2.3% which is not significant (*p* ≥ 0.001) as compared with ascorbic acid (89.3 ± 2.1%) as a positive control (Figure 5A). Additionally, this finding was recorded for a concentration of 100 µg mL^−1^, the antioxidant activity between Ag-NPs and ascorbic acid was not significant with scavenging percentages of 84.0 ± 1.5% and 85.1 ± 1.1%, respectively. The lowest scavenging activity was recorded for a concentration of 0.78 µg mL^−1^ with a percentage of 6.2 ± 1.1% followed by a concentration of 1.56 µg mL^−1^ with a percentage of 33.5 ± 1.5% as compared to ascorbic acid at the same concentrations (10.8 ±2.3% and 40.6 ± 2.7%, respectively) (Figure 5A). The antioxidant activity of Ag-NPs fabricated by *Psidium guajava* water extract was assessed by DPPH and did not show any significant differences at concentrations 100 and 120 µg mL^−1^ as compared to the control (ascorbic acid) [23]. The authors reported that the scavenging percentages due to 100 and 120 µg mL^−1^ of Ag-NPs were 83.6% and 89.0%, respectively as compared to ascorbic acid showed scavenging percentages of 89.0% and 90% at the same concentrations. Interestingly, the results of our investigation (83.9%) were higher than those recorded by Ravichandran et al., who found that Ag-NPs manufactured using the leaf extract of *Atrocarpus altilis* had radical scavenging capabilities for DPPH of 79.79% [22].

The H_2_O_2_ scavenging assay method (Figure 5B) was matched with the DPPH assay method in the concentration-dependent phenomenon, but it differs in that there are significant differences between the scavenging activity of Ag-NPs and ascorbic acid in all treatments. As shown, the highest Ag-NPs scavenging activity was recorded for a concentration of 200 µg mL^−1^ with percentages of 76.5 ± 1.7%, and the lowest activity was showed at a concentration of 0.78 with percentages of 5.3 ± 1.1% as compared with ascorbic acid (81.6 ± 1.3% and 11.3 ± 2.7% for the same concentrations, respectively). At a concentration of 100 µg mL^−1^, our findings were in line with those that previously indicated that the H_2_O_2_ scavenging activity for Ag-NPs produced using the aqueous extract of *Calophyllum tomentosum’s* leaves was 83.9% and that of ascorbic acid was 79.7% [17].

The concentration of active compounds to scavenging 50% is defined as effective concentration (EC_50_). The detection of EC50 is an important factor for investigating and comparing the antioxidant activity of different molecules, optimizing therapeutic concentrations, safety assessment, and cost-effectiveness for the development of a new drug [57]. Herein, the EC50 of green synthesized Ag-NPs and ascorbic acid were (10.4 and 5.7 µg mL^−1^) and (23.9 and 11.9 µg mL^−1^) for DPPH and H_2_O_2_ assay methods, respectively. Recently, the EC50 value of Ag-NPs formed using *Erythrina suberosa* was 30.04 µg mL^−1^ based on the DPPH method [21].

The antioxidant activity of fabricated NPs using biological methods, especially plant extracts, is better than the activity of those synthesized by chemical and physical methods. It is possible that this phenomenon is due to the capping agent that is derived from the plant extract. This capping agent includes phenolic chemicals, flavonoids, alkaloids, and polysaccharides, all of which enhance and multiply the antioxidant capacity of NPs [23,24]. The antioxidant potential of Ag-NPs can be explained by their ability to quench or scavenge free radicals by donating or accepting electrons. The reason behind this is that, depending on the circumstances of the reaction, silver can exist in two oxidation states: Ag^+^ and Ag^2+^ [58]. Overall, the green synthesized Ag-NPs can be utilized as an antioxidant agent for safe protecting human health against degenerative disease caused by various oxidative stresses.

### 2.5. Antiviral Activity

In this investigation, the efficacy of *P. oleracea*-based Ag-NPs to fight the growth of HAV and Coxsackie B virus was evaluated. First, the cytotoxic potential of Ag-NPs against normal Vero cell lines, as host cells, was investigated to detect the MNTC which is 250 µg mL^−1^. In another investigation, the MNTC value of Ag-NPs formed using an aqueous extract of *Lampranthus coccineus* and *Malephora lutea* was detected by investigating the cytotoxic efficacy against Vero cells [19]. The authors reported that the MNTC was 260.3 and 520.6 and µg mL^−1^ for Ag-NPs produced by *M. lutea* and *L. coccineus*, respectively.

The MTT antiviral assay method showed that the viability of Vero cells inoculated with viruses and treated with Ag-NPs was increased from (34.25 ± 1.4% and 23.85 ± 0.5%) to (86.3 ± 0.8% and 79.89 ± 0.7%) for HAV and Cox-B4, respectively (Figure 6). The obtained results mean the Vero cell viability after being treated with HAV or Cox-B4 viruses in the presence of green synthesized Ag-NPs was increased with the value of 2.52- and 3.35-fold as compared with the viability in the presence of viruses and absence of NPs. On the other hand, data analysis reveals that the as-formed Ag-NPs have antiviral effects with percentages of 79.16% and 73.59% for HAV and Cox-B4, respectively (Figure 6).

The antiviral efficacy of NPs can be attributed to different mechanisms such as the interaction between viruses and cells leading to prohibiting the infection, preventing the entrance of viruses to the host cells through the interact of NPs with cell surfaces or specific receptors, inhibiting the viruses’ replications, prevent the spread of viruses, enhance the oxidative stress through production of ROS, cell apoptosis, and enhance the immune response of the host cells [59,60]. The ability of Ag-NPs to block virus attachment to host cells and hence cell damage is achieved by their interaction with the viral envelope, receptors, or surface proteins. NPs have the potential to block virus replications inside the host cells by the function of nucleic acids (DNA or RNA) and blocking the essential enzymes needed for virus replications [61]. In addition, when exposed to NPs, infected cells may release ROS which has damage effects on viral DNA, proteins, and lipid membranes, ultimately hindering the virus’s capacity to multiply and infect cells [62]. It is also possible for Ag-NPs to influence the host’s immunological response and boost the host’s immunity by increasing the synthesis of interferons and proinflammatory cytokines that are used to identify and react to viral infections [63]. With the help of this enhanced immune response, infected cells can be eliminated, and the spread of viruses can be controlled. Ag-NPs can hinder the apoptotic pathways and hence give more time to the immune system to combat the pathogens [63]. Some authors reported that the antiviral activity of green synthesized NPs can be affected by coating agents such as flavonoids, alkaloids, and terpenoids that enhance the antiviral potential [64].

## 3. Materials and Methods

### 3.1. Materials

To synthesize Ag-NPs in an eco-friendly manner, analytical grade (99%) silver nitrate (AgNO_3_) obtained from Sigma Aldrich (Darmstadt, Germany) was used. For the antimicrobial assay, coded test strains were obtained from the ATCC. Cox-B4 and HAV, obtained from the Holding Company for Biological Products and Vaccines (VACSERA) in Doki, Giza, Egypt, were the viruses tested in the anti-viral assay.

### 3.2. Plant-Synthesized Ag-NPs

The *P. oleracea* plant leaves were collected, washed with tap water, and left to dry and then grinded to form powder. Following this, the formed powder was mixed with distilled H_2_O (5 g/100 mL) and subjected to heating for 60 min at 40 °C under agitation conditions (150 rpm). To obtain the supernatant for the production of Ag-NPs, the mixture was centrifuged at 10,000 rpm for 7 min. For biosynthesis, AgNO_3_ (16.9 mg) was mixed with 80 mL dH_2_O followed by adding 20 mL of plant water extract to achieve a final concentration of 1 mM of Ag-NPs. The previous solution underwent stirring conditions for one hour at 150 rpm and the pH of the mixture was 8 by adding NaOH (1 M) drop-by-drop. After that, the solution remained at room temperature overnight to confirm a complete reduction of metal precursor to form a nanostructure which was detected by color change from colorless to yellowish brown.

### 3.3. Characterization

To detect the surface plasmon resonance (SPR) of the produced Ag-NPs, ultraviolet-visible spectroscopy (JENWAY 6305, Cole-Parmer, Stafford, UK) was used to measure the absorbance of the resultant color from 200 to 600 nm. Furthermore, the plant water extract’s functional groups and their contributions to the bio-reduction of AgNO_3_ to Ag-NPs were determined using Fourier transform infrared (FT-IR, Cary 630, Tokyo, Japan). Here, a disk containing 300 mg of the Ag-NPs was compressed into a mixture of KBr and scanned over a broad spectrum of wavenumbers (400–4000 cm^−1^) [7]. Transmission electron microscopy (TEM, JEOL 1010, Tokyo, Japan) was used to characterize the size and shape of the NPs. The carbon/copper TEM grid was filled with the NPs solution until full adsorption occurred. Before analysis, gent touched of the loaded grid with blotting paper was performed to remove any surplus solution. Energy-dispersive X-ray (EDX, JEOL, JSM6360LA, Tokyo, Japan) was used to map the elemental composition of the produced Ag-NPs [65].

An X-ray diffraction pattern (XRD, X’ Pert Pro, Philips, Eindhoven, The Netherlands) was employed to investigate the crystalline structure of the Ag-NPs that were produced. Under the settings of 40 kV voltage, 30 mA current, and a Cu-Kα radiation source with a λ value of 1.54 Å, the 2θ values of the XRD pattern were observed within the range of 5–80°. In addition, the average crystallite size of the Ag-NPs was calculated using the Debye–Scherrer equation utilized during the XRD investigation as follows:(1)$$D=\frac{0.9\;\times \;1.54}{\beta \cos\theta }$$
where 0.9 is the constant of Scherrer’s, 1.54 is the radiation value, *β* is half of the maximum intensity, and θ is the Bragg’s angle.

Zeta potential (ζ) analysis and dynamic light scattering (DLS) were used to determine the surface charge and size distribution of Ag-NPs in the colloidal solution, respectively. To avoid any shadows from appearing during the examination, the produced Ag-NPs were dissolved in high pure water [47]. The surface charge of the Ag-NPs was measured using the Malvern Zetasizer, an instrument manufactured by Nano-ZS and located in Malvern, UK.

### 3.4. Pharmacological Activities

#### 3.4.1. Antimicrobial Activity

Different prokaryotic pathogenic strains designated as G+ve bacteria (*Staphylococcus aureus* ATCC6528 and *Bacillus subtilis* ATCC6533) and G-ve bacteria (*Pseudomonas aeruginosa* ATCC9027 and *Escherichia coli* ATCC8739) as well as eukaryotic strains represented by unicellular fungi (*Candida albicans* ATCC10231), and multicellular fungi *(Aspergillus brasiliensis* ATCC16404), were employed to investigate the antimicrobial potential of plant-based Ag-NPs. The agar well diffusion method was used to investigate this efficacy [66]. At first, nutrient agar, sabouraud dextrose agar, and Capek’s Dox agar plates were used to refresh the growth of selected bacterial, yeast, and *Aspergillus* strains, respectively. The inoculated plates were put in an incubator for 24 h at 35 ± 2 °C/or 72 h at 28 °C for bacteria/yeast and *A. brasiliensis,* respectively. After that, the newly cultured bacterial and yeast strains were reinoculated on the surface of plates containing Muller–Hinton agar media for bacteria and yeast, and Capek’s Dox agar media for *Aspergillus* strain. Using a sterile cork borer, four wells of 0.6 mm in diameter were made on each agar plate. After that, 100 µL of various Ag-NP concentrations (200, 100, 50, 25, and 12.5 µg mL^−1^) were added to these wells. The plates were then placed in the refrigerator for an hour before being incubated at a temperature of 35 ± 2 °C for 24 h. A well was then treated with DMSO (a solvent system, ≥ 99.7%, Merck, Darmstadt, Germany) to serve as a control. As part of the data collection process, the diameter of the clear zone around each well was measured in millimeters (mm). A distinct zone surrounding the well-formed by the lowest concentration of green-synthesized Ag-NPs was used to determine the minimum inhibitory concentration (MIC) value. The experiment was achieved in triplicate.

#### 3.4.2. Antioxidant Activity

##### DPPH Assay Method

The DPPH (2,2-diphenyl-1-picrylhydrazyl (2,2-diphenyl-1-picrylhydrazyl) technique was utilized to evaluate the antioxidant potential of produced Ag-NPs. By employing this particular technique, a range of quantities (ranging from 0.78 to 200 µg mL^−1^) of Ag-NPs were prepared in Milli-Q H_2_O. A test tube containing one mL of DPPH (dissolved in methanol), 450 µL of Tris-HCl buffer (pH of 7.4), and one mL of the Ag-NPs solution has been made to achieve the analysis. The mixture was thoroughly mixed and put in an incubator for 30 min., at 37 °C, and with a shaken condition of 100 rpm in the dark condition. The same conditions and concentrations were used in an additional series of experiments with ascorbic acid serving as the positive control. In the same experimental setup, a test tube containing DPPH and tris-buffer in the absence of any treatment (Ag-NPs or ascorbic acid) was used as a negative control. At the end of the incubation period, the absorbance of the generated color was determined at 517 nm. To determine the free radical scavenging percentages, the following formulae were utilized [57]:(2)$$\mathrm{DPPH}\;\mathrm{scavenging}\;\%=\frac{\mathrm{absorbance}\;\mathrm{of}\;\mathrm{control}-\mathrm{absorbance}\;\mathrm{of}\;\mathrm{treatment}}{\mathrm{absorbance}\;\mathrm{of}\;\mathrm{control}}\times 100$$

##### H_2_O_2_ Assay Method

The antioxidant efficacy of formed Ag-NPs was also evaluated using hydrogen peroxide [H_2_O_2_, 10 mM, dissolved in phosphate buffer (pH = 7.4 and 0.1 M)]. Different Ag-NP concentrations were prepared as follows: 0.78 to 200 µg mL^−1^. In this method, one mL of Ag-NP solution was mixed well with 2 mL of the prepared H_2_O_2_ before incubated for 10 min at 37 °C. The JENWAY 6305 spectrophotometer was used to monitor the absorbance of the produced color at 230 nm. The experiment was conducted with the control, which was an Ag-NPs solution devoid of H_2_O_2_. The following formula was used to compute the H_2_O_2_ scavenging activity:(3)$$H_{2}O_{2}\mathrm{scavenging}\;\%=\frac{A-B}{A}\times 100$$
where A and B were the absorbances of control and treatment, respectively.

#### 3.4.3. Antiviral Activity

##### Detection of Maximum Non-Toxic Concentration (MNTC) of Biosynthesized Ag-NPs

The biosynthesized Ag-NPs were integrated into a 96-well plate with a normal Vero cell which displayed as host cells to calculate the MNTC using MTT assay. To create a confluent sheet of Vero cells, 10^4^ cells/well in 100 µL growth media were cultured in a CO_2_ incubator (5%) at 37 °C for 24 h. Post-growth media removal before being treated the Vero cells with Ag-NPs concentrations (1000, 500, 250, 125, 62.5, and 31.25 µg mL^−1^). DMSO-dissolved generated NPs were tested in triplicate at each concentration. Untreated cells were used as the control. The treated plates were kept in a CO_2_ incubator for 48 h at 37 °C in a 5%. Afterward, 20 µL of MTT solution (5 mg mL^−1^/phosphate buffer) was added to each well, stirred for 5 min at 150 rpm, and then placed in a CO_2_ incubator at 37 °C for 5 h. Final step: the well contents were discarded, and 200 µL DMSO were added per well to dissolve formazan crystals (MTT metabolic product). ELIZA microplate readers were used to measure the treated plates’ OD at 570 nm.

The viability of the cells was measured using the next equation:(4)$$Cell\;viability\;(\%)=\frac{A}{B}\times \;100$$
where *A* and *B* are the absorbance of treatment and control, respectively.

On the other hand, the toxicity of the cells was measured using the next equation:
(5)Cell toxicity (%) = cell viability − 100

Then, the MNTC was calculated by graphing the relationship between the toxicity percentages and various Ag-NP concentrations. Additionally, the CC50, which is defined as the Ag-NP concentration needed for 50% growth inhibition of Vero cells was measured by regression analysis.

##### Antiviral Activity

Ag-NPs were tested for antiviral activity against the hepatitis A virus (HAV) and Coxsackie B virus. First, host Vero cells were grown into a confluent sheet. In parallel, the viral suspension was incubated with non-toxic NPs (1:1, *v*/*v*) at 35 ± 2 °C for one hour. Finally, 100 µL of the virus-Ag-NPs mixture was applied to each Vero cell well. Three independent wells with only Vero cells represented the control (non-infected cells). Shaking for 5 min at 150 rpm was performed to mix the content of the plate, which was then incubated overnight at 37 °C with 5% CO_2_. After applying the MTT solution, formazan crystal absorbance was measured to determine Vero cell viability. The antiviral activity was determined by comparing the optical density of infected and uninfected cellular viabilities [67].

### 3.5. Data Analysis

Statistical analysis was performed on the collected data using SPSS version 18 (SPSS Inc., Chicago, IL, USA). Before using Tukey’s multiple comparison test, an analysis of variance (ANOVA) was performed to compare different samples.

## 4. Conclusions

The “green synthesis” of Ag-NPs from the *P. oleracea* water extract provides an environmentally friendly and efficient process for safe NP production. The morphological study confirms the presence of homogeneous Ag-NPs with 5–40 nm sizes. Silver plays a dominant role, according to EDX. The XRD confirms that they are crystals. The effect of capping agents on dynamic light scattering reveals a bigger hydrodynamic diameter. The results of the zeta potential test show that the material is very stable and has a negative surface charge. The formed Ag-NPs are particularly effective against G-ve bacteria, the antibacterial action of the Ag-NPs is concentration-dependent. Furthermore, they can scavenge free radicals and show antioxidant characteristics. The antiviral effects of Ag-NPs against hepatitis A and Coxsackie B viruses are also encouraging. While these findings are promising, the current study needs more investigation to explore the role of plant active metabolites in the fabrication of Ag-NPs. The antimicrobial activity against wide range of multicellular fungi needs more studies. In addition, other methods to clarify the antioxidant activity and investigate their antiviral against other human viruses need more explanations.

## Figures and Tables

**Figure 1 pharmaceuticals-17-00317-f001:**
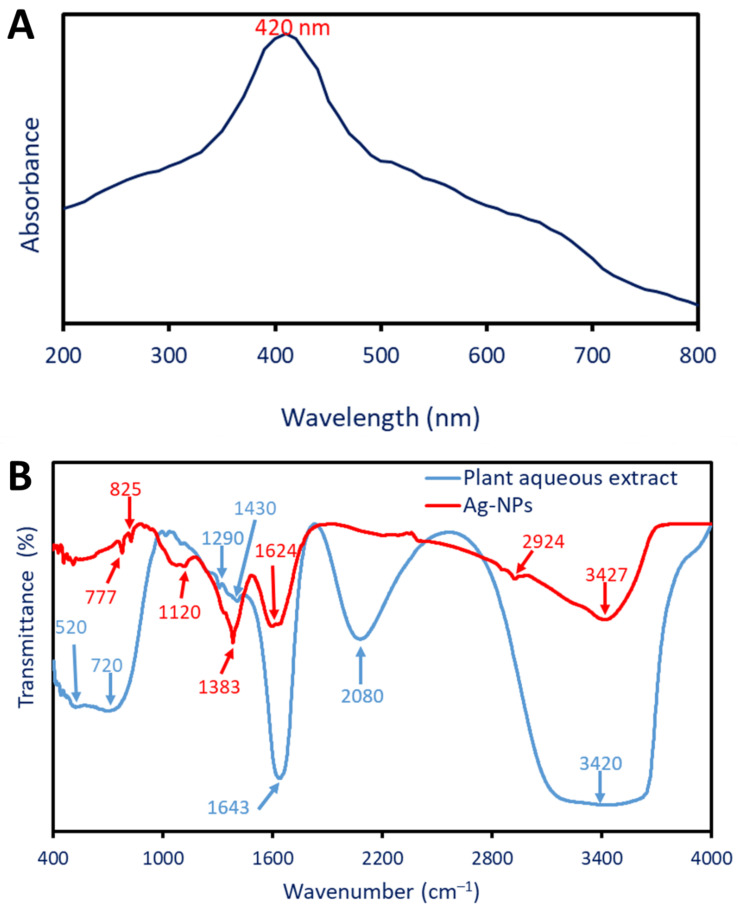
Green synthesized Ag-NPs exhibit maximal SPR at 420 nm in their optical characteristics (**A**), and a variety of functional groups in both the plant aqueous extract and the Ag-NPs (**B**) as shown by FT-IR.

**Figure 2 pharmaceuticals-17-00317-f002:**
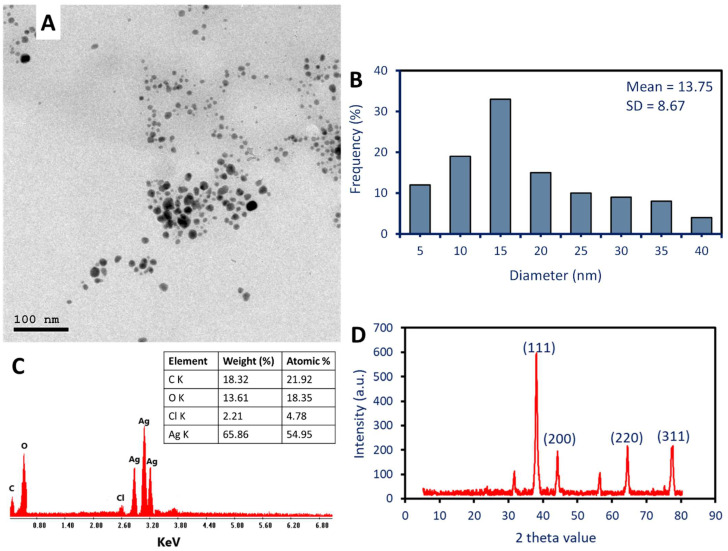
Study of Ag-NPs Profile. The spherical and well-arranged shape of the Ag-NPs is shown in (**A**), the size distribution in (**B**), the composition of the as-formed Ag-NPs is shown in (**C**), and the crystalline nature of the Ag-NPs is validated in (**D**) through XRD.

**Figure 3 pharmaceuticals-17-00317-f003:**
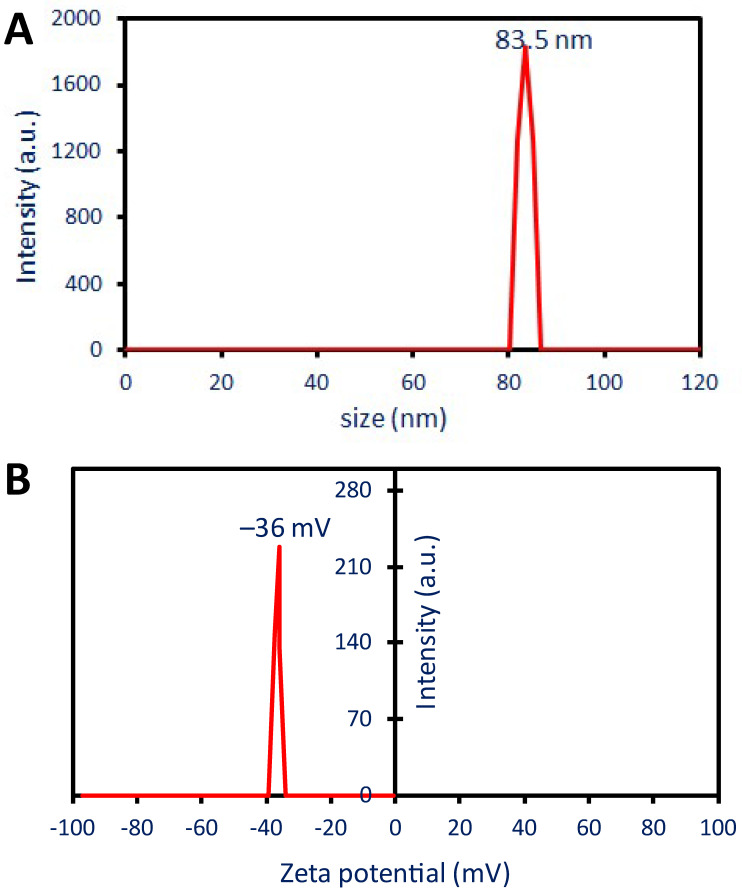
The stability and surface charge of produced Ag-NPs as assessed by dynamic light scattering (**A**), and zeta potential (**B**).

**Figure 4 pharmaceuticals-17-00317-f004:**
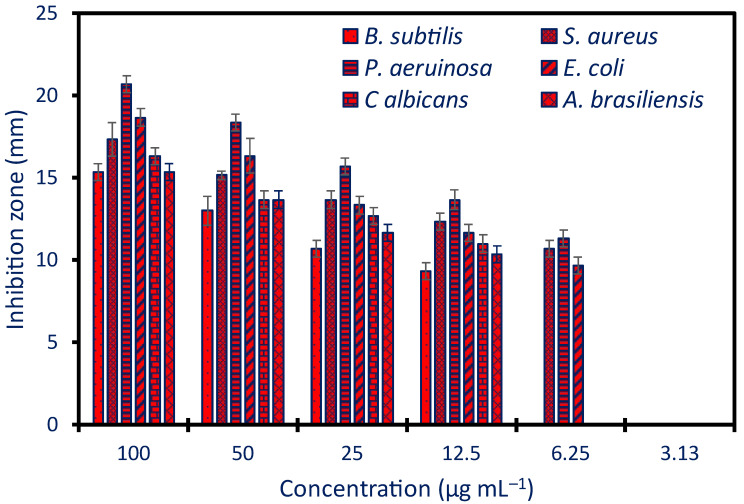
The antimicrobial efficacy of Ag-NPs derived from plants against G+ve bacteria (*B. subtilis* and *S. aureus*), G-ve bacteria (*P. aeruginosa* and *E. coli*), as well as unicellular (*C. albicans*) and multicellular (*A. brasiliensis*) fungi.

**Figure 5 pharmaceuticals-17-00317-f005:**
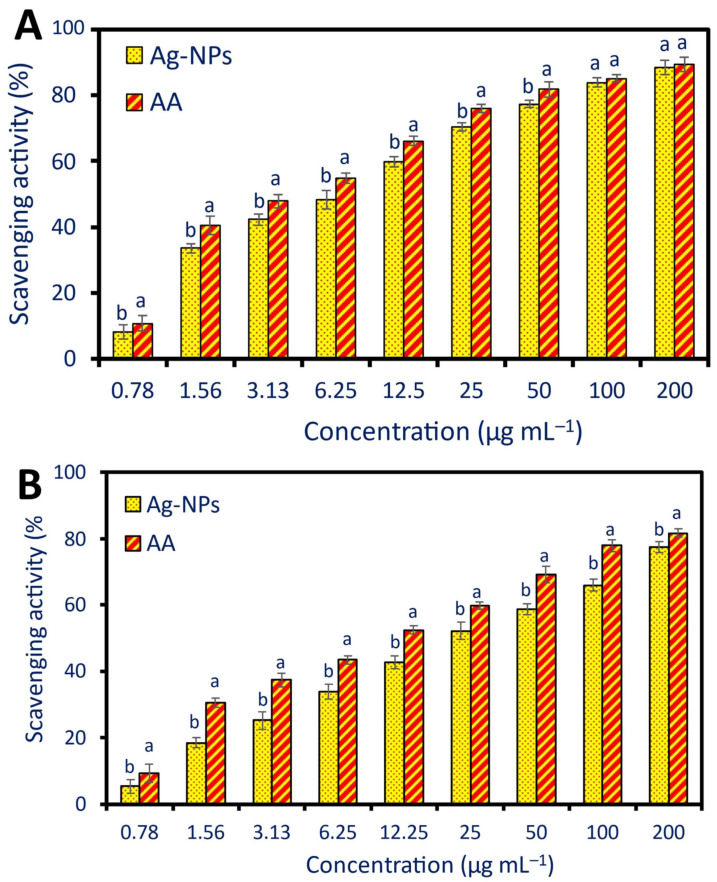
Antioxidant activity of plant-based Ag-NPs assessed using DPPH assay method (**A**) and H_2_O_2_ assay method (**B**). AA denotes the ascorbic acid (control). Different letters in the same concentration indicate the data are significant (*p* ≤ 0.05).

**Figure 6 pharmaceuticals-17-00317-f006:**
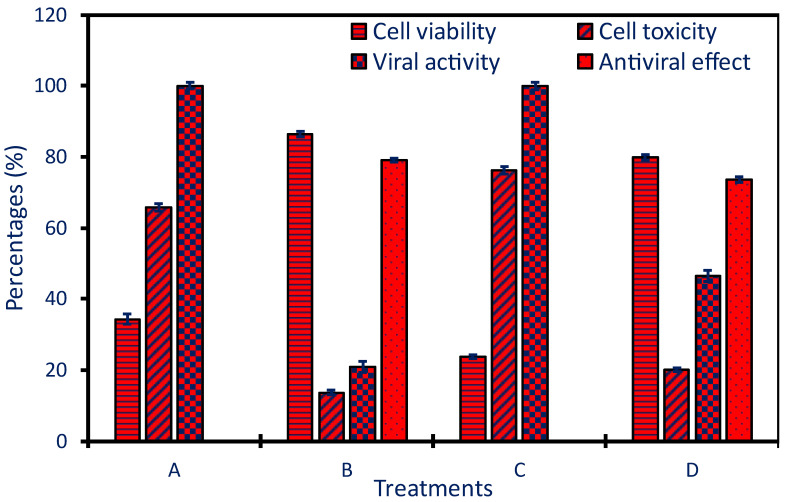
Hepatitis A (HAV) and Cox-B4 virus antiviral activities of Ag-NPs produced using plant extract. (**A**) is the Vero host cells treated with HAV, (**B**) is Vero cells treated with HAV in the presence of Ag-NPs, (**C**) is the Vero cells treated with Cox-B4, and (**D**) is the Vero cells inoculated with Cox-B4 in the presence of Ag-NPs.

**Table 1 pharmaceuticals-17-00317-t001:** The MIC values with inhibition clear zones of Ag-NPs against different test organisms.

Test Organism	MIC Value (µg mL^−1^)	Clear Zone (mm)
*B. subtilis*	12.5	9.3 ± 0.5
*S. aureus*	6.25	10.7 ± 0.6
*P. aeruginosa*	6.25	11.3 ± 0.5
*E. coli*	6.25	9.7 ± 0.5
*C. albicans*	12.5	11.0 ± 0.5
*A. brasiliensis*	12.5	10.3 ± 0.5

## Data Availability

The data presented in this study are available on request from the corresponding authors.

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
