# Peer review of "Exploring the Antimicrobial, Antioxidant, and Antiviral Potential of Eco-Friendly Synthesized Silver Nanoparticles Using Leaf Aqueous Extract of Portulaca oleracea L."

_pharmaceuticals, 2024, doi:10.3390/ph17030317_

Round 1

Reviewer 1 Report

Comments and Suggestions for Authors

The development of nanotechnology has introduced nanomaterials into the medical device sector. The most frequent applications in the biomedical field are in the diagnostic and therapeutic fields. Silver nanoparticles are widely used for biomedical applications where antibacterial, antifungal, antiviral properties, etc. are exploited. In particular, the use of environmentally friendly materials such as medicinal plant matrix extract for the synthesis of silver nanoparticles offers numerous advantages in terms of eco-sustainability and compatibility with pharmaceutical and biomedical applications, since it does not use toxic chemicals for the synthesis protocol. Green synthesis represents a step forward compared to chemical and physical methods as it is cost-effective and environmentally friendly. In this work, the potential of an aqueous extract of Portulaca oleracea to assemble silver nanoparticles (Ag-NPs) for use in different biomedical applications was evaluated. Therefore, Ag-NPs were characterized using different analytical tools, and the activity of Ag-NPs as an antimicrobial and antiviral agent was investigated.

The topic covered in this work is very current. The methods have been adequately described. The results are well presented and the interpretation of the data is appropriate. The discussion is sufficiently elaborate, with a clear conclusion. The literature is well represented and sufficiently updated.

However, some changes are required, as follows.

Lines 32-35: Reformulate the sentence less drastically, as the possible use of P. oleracea-based silver particles for human health requires further investigation.

Lines 376-378: Restate the sentence, as explained above.

In the conclusion section, insert a brief sentence on the limitations of the study and future prospects.

Comments on the Quality of English Language

The manuscript is easily readable as far as the English language is concerned.

Author Response

Thank you for your comments. We answer all comments point-by-point as shown in the file attachment.

Reviewer 2 Report

Comments and Suggestions for Authors

In the present manuscript Abdel-Rahman etal., synthesized silver nanoparticles using aqueous leaf extract of Portulaca oleracea and evaluated their antimicrobial, antioxidant and antiviral potential. The manuscript is very nice and the authors presented a solid study.  I appreciate the authors for such an elaborate and detailed results and discussion section. The following points should be considered.

1. why did the authors not use any positive controls in the antimicrobial study?

2. Authors should justify the novelty of the study in the introduction. As literature reported few papers on the silver nanoparticles synthesized by the leaf extract of Portulaca oleracea.

3. Please provide the MIC and MBC values of Portulaca oleracea AgNPs  Against the tested pathogens in tabular form. 

4. Please provide the statistical significance data in figure 5. 

Author Response

(The authors gave the same response as above.)

Reviewer 3 Report

Comments and Suggestions for Authors

The paper investigates the green fabrication of silver nanoparticles (Ag-NPs) using Portulaca oleracea leaf extract and explores their biomedical applications. Characterization techniques, including UV-Vis spectroscopy, TEM, EDX, and XRD, confirm the successful synthesis and properties of Ag-NPs. The synthesized Ag-NPs exhibit promising antimicrobial, antioxidant, and antiviral activities. However, the review lacks clarity and conciseness, requiring improved organization and focus on key findings.

  1. Could the introduction be more succinct in presenting the unique contribution of this study in the context of green-synthesized Ag-NPs for biomedical applications?
  2. Can the authors clarify the rationale behind choosing P. oleracea for Ag-NP synthesis and highlight its advantages over other plant extracts?
  3. Are there additional insights on the potential mechanisms underlying the superior antimicrobial activity against Gram-negative bacteria compared to Gram-positive bacteria?
  4. How could the discussion better emphasize the practical implications and future directions for utilizing P. oleracea-based Ag-NPs in real-world biomedical scenarios?
  5. Can the authors provide more details on the experimental conditions and controls employed in antioxidant assays to enhance the reliability of the reported scavenging percentages?

Author Response

(The authors gave the same response as above.)

Reviewer 4 Report

Comments and Suggestions for Authors

Minor issues 

- Please consider to provide some background information in the abstract.

- Line 56: “plants are characterized by their huge metabolites…”. This sentence is confusing. Maybe “huge amount of metabolites” is more appropriate.

- Line 61: Please provide the full specie name of Portulaca oleracea at the first citation as indicated at “The World Flora online” (https://www.worldfloraonline.org/)

- Line 65: huge phyto-constituent?

- Please attempt to use the abbreviated form of species names after the first citation. 
- Lines 278-282: The authors should perform statistical analysis for prove the most active concentration. This analysis should be represented in the figure 4. 
- Figure 5: please indicate the meaning of the superscript letters in the legend.

- Figure 6: please indicate if the results was analyzed by any statistical method.

- Line 472: please consider to change “Biological activity” for “pharmacological activities”

Author Response

(The authors gave the same response as above.)
